# A New Genus of Spittlebugs (Hemiptera, Cercopidae) from the Eocene of Central Tibetan Plateau [note 1]

**DOI:** 10.3390/insects13090770

**Published:** 2022-08-25

**Authors:** Xiao-Ting Xu, Jacek Szwedo, Di-Ying Huang, Wei-Yu-Dong Deng, Martyna Obroślak, Fei-Xiang Wu, Tao Su

**Affiliations:** 1CAS Key Laboratory of Tropical Forest Ecology, Xishuangbanna Tropical Botanical Garden, Chinese Academy of Sciences, Mengla 666303, China; 2University of Chinese Academy of Sciences, Beijing 100049, China; 3Laboratory of Evolutionary Entomology and Museum of Amber Inclusions, Department of Invertebrate Zoology and Parasitology, University of Gdańsk, 59, Wita Stwosza Street, PL80-308 Gdańsk, Poland; 4State Key Laboratory of Palaeobiology and Stratigraphy, Center for Excellence in Life and Paleoenvironment, Nanjing Institute of Geology and Palaeontology, Chinese Academy of Sciences, Nanjing 210008, China; 5Hessisches Landesmuseum Darmstadt, 64283 Darmstadt, Germany; 6Key Laboratory of Vertebrate Evolution and Human Origins of Chinese Academy of Sciences, Institute of Vertebrate Paleontology and Paleoanthropology, Chinese Academy of Sciences, Beijing 100044, China

**Keywords:** Insecta, Cercopidae, *Nangamostethos tibetense*, fossil, late Eocene, new genus, new species, Tibetan Plateau, Niubao Formation

## Abstract

**Simple Summary:**

The superfamily Cercopoidea is commonly named as “spittlebugs”, as its nymphs excrete a spittle mass to protect themselves and hide from predators and parasites. Cosmoscartini (Cercopoidea: Cercopidae) is a large and brightly colored Old World tropical tribe, including 11 genera. A new genus *Nangamostethos* gen. nov. (type species: *Nangamostethos*
*tibetense* sp. nov.) of Cosmoscartini is described from Niubao Formation, the late Eocene (ca. 39 million years ago) of central Tibetan Plateau (TP), China. Its placement is ensured by comparison with all the extant genera of the tribe Cosmoscartini. The new fossil represents one of the few fossil Cercopidae species described from Asia. It is likely that *Nangamostethos* was extinct from the TP due to the regional aridification and an overturn of plant taxa in the last 33 million years.

**Abstract:**

The superfamily Cercopoidea is commonly named as “spittlebugs”, as its nymphs produce a spittle mass to protect themselves. Cosmoscartini (Cercopoidea: Cercopidae) is a large and brightly colored Old World tropical tribe, including 11 genera. A new genus *Nangamostethos* gen. nov. (type species: *Nangamostethos*
*tibetense* sp. nov.) of Cosmoscartini is described from Niubao Formation, the late Eocene of central Tibetan Plateau (TP), China. Its placement is ensured by comparison with all the extant genera of the tribe Cosmoscartini. The new fossil represents one of few fossil Cercopidae species described from Asia. It is likely that *Nangamostethos* was extinct from the TP due to the regional aridification and an overturn of plant taxa in the late Paleogene.

## 1. Introduction

The members of the superfamily Cercopoidea (Hemiptera: Cicadomorpha), commonly known as spittlebugs, are characterized by the conical and short hind coxa, body and wings clothed in fine setae, and the presence of one or two lateral spines and two rows of apical spines on the hind tibia [1,2]. Although spittlebugs are widespread all around the world, their species diversity centers are in the tropics, where approximately 70% of the described species can be found [3].

Cercopoidea comprises five extant families (Cercopidae, Aphrophoridae, Epipygidae, Clastopteridae, and Machaerotidae) and three extinct ones (Procercopidae, Sinoalidae, and Cercopionidae) [4,5]. Few comprehensive phylogenetic analyses have been attempted to date [3,6], which means that the monophyly of these families remains equivocal [3,5]. The family Cercopidae includes 178 genera and 1561 species [7], most closely related to the paraphyletic Aphrophoridae [6], which bear transverse compound eyes instead of the globose compound eyes of Cercopidae [8]. The fossil record of Cercopidae is large, with more than 40 species reported from the Paleocene to the Pleistocene in Eurasia and the Americas [9,10]. However, most of the record was attributed to the genera or even the family with relatively weak support. The reliable record of Cercopidae is more limited, and therefore needs a further revision. The oldest confirmed fossil known for the Cercopidae (*Allocercopis punctatis* Lin, 1997) dates back to the Cretaceous–Paleocene [11] and two dubious cercopid fossils were recorded from the Cretaceous [12,13].

Tegmen patterns in the subfamily Cercopinae often include red, orange, or yellow markings on a black background [14]. Those striking color patterns appear to represent warning coloration, which probably in conjunction with reflex bleeding (exuding hemolymph from rupture lines in pretarsal pads when attacked) acts to deter predators [15].

Most spittlebugs are oligo- or polyphagous [5,14,16]. Cercopids are xylem feeders and have an enlarged postclypeus to ingest plant fluids under negative pressure [5]. A high proportion of cercopids, in regard to abundance and species diversity, show a statistically significant preference for nitrogen-fixing host plants [17].

Herein, we describe a new genus *Nangamostethos* gen. nov. from the upper Eocene Niubao Formation at Kanggale Hill, Nima Basin, in the central Tibetan Plateau, China.

## 2. Materials and Methods

The three specimens of fossil cercopid in this study are preserved as carbon impressions on the surface of shales, collected from the Xiede section, northern Kanggale Hill, Nima Basin (31°58′23″N, 88°25′42″E, 4662 m a.m.s.l.; Figure 1).

The Nima Basin is a Cenozoic continental basin in the central Tibetan Plateau [19]. It is an east–west-trending elongated basin with an area of approximated 3000 km^2^, situated in the Bangong–Nujiang suture zone and formed by the Mesozoic collision of the Lhasa and Qiangtang terranes [19,20,21]. The fossil-bearing strata comprise greyish-green mudstones and calcareous shales, interbedded with mudstones, sandstones, and limestones [22]. The Nima Basin is located to the west of the adjacent Lunpola Basin [19] (Figure 1A). According to the latest chronostratigraphic work with radiochronologic dating methods, the age of the fossil-bearing layer in Dayu, Lunpola Basin is ∼39 Ma [23]. We consider that Xiede and Dayu sites are age equivalent as the two localities share similar stratigraphic sequences, deposition fabric, and a wide range of fossils including insects, plant macrofossils, and vertebrates (e.g., *Eoanabas*, Wu et al., 2017 [24]; *Cedrelospermum*, Jia et al., 2018 [25]; *Aquarius*, Cai et al., 2019 [26], and *Sabalites*, Su et al., 2019 [27]).

The specimens were observed under a Leica S8AP0 stereomicroscope, and photographs were taken with a Canon 77D digital camera, a digital camera attached to a Zeiss Discovery V16 microscope, and a Smartzoom 5 digital camera. Specimens were immersed in 70% alcohol, which improves the contrast between the image and matrix and reveals more morphological details. The fossil insects were housed in the Paleoecology Collections, Xishuangbanna Tropical Botanical Garden, Chinese Academy of Sciences, Mengla, China.

We followed the nomenclature of wing venation for all Paraneoptera [28] and Wang et al. [29] for the terminologies of the cercopoid head.

## 3. Results


*Systematics*


Family Cercopidae, Leach, 1815

Subfamily Cercopinae Leach, 1815

Tribe Cosmoscartini Schmidt, 1920

Genus *Nangamostethos* Xu et Szwedo, gen. nov.

urn:lsid:zoobank.org:act:77C90479-B171-4D10-BFB6-5A55026FB0FA

Type species *Nangamostethos tibetense* Xu et Szwedo, sp. nov.; here designated.

Etymology. From Tibetan ‘na ngamo’ (གནའ་སྔ་མོ in THL Simplified Phonetic Transcription; gna’ snga mo in Wylie transliteration) meaning ancient and Classic Greek ‘stethos’ (στήθος) meaning thorax. Gender: neuter.

Diagnosis. Externally similar to some species of *Cosmoscarta* Stål, 1869 and *Phymatostetha* Stål, 1870 by size, enlarged pronotum, and coloration with two distinct pale transverse bands and two pale spots at the base of corium and clavus. It differs from both genera by more basal branching of stem ScP + R (in *Cosmoscarta* and *Phymatostetha* stem ScP + R is forked more close to half of corium length); more clear part of CuA_1_ branch reaching branch MP_3 + 4_ at the border of corium and membrane (reticulate branches in *Cosmoscarta* and *Phymatostetha*); membrane with less distinct reticulate branches (distinct, irregular branches on membrane in *Phymatostetha* and *Cosmoscarta*); anterior margin of head in dorsal view with more angulate border of frontoclypeus (rounded and strongly produced in *Phymatostetha* and *Cosmoscarta*); median carina of frontoclypeus absent (as in *Phymatostetha* and *Cosmoscarta*); pronotum with median expansion at anterior margin (present in some *Phymatostetha* but less distinct); lateral angles of pronotum distinct (similar or even more expanded in *Phymatostetha*, less expanded and more rounded in *Cosmoscarta*); posterior margin of pronotum with distinct median incision (present but more shallow in *Phymatostetha* and absent in some *Cosmoscarta*); distinctly longer mesonotum, about 0.8–0.9 as long as pronotum in mid line (mesonotum about 0.5 times as long as pronotum in *Phymatostetha* and *Cosmoscarta*).

*Nangamostethos tibetense* Xu et Szwedo, sp. nov.

Figure 2, Figure 3, Figure 4 and Figure 5.

urn:lsid:zoobank.org:act:26CA2296-D3C4-49B6-839B-EE72ADA1DA65

Etymology. Named after the Tibetan region where the type was found.

Materials. Holotype, specimen XDB3-1209 (a compression of a nearly complete adult, but with venation of the tegmen poorly preserved) and paratypes XDB3-0252, XDA2-0430 (part and counterpart) housed in the Xishuangbanna Tropical Botanical Garden, Chinese Academy of Sciences, Mengla, China (XTBG).

Diagnosis. Body 11.10–12.03 mm long without tegmen; three longitudinal pale stripes on dark brown pronotum and scutellum; legs pale with dark banding; four oval or band-shaped pale markings on each dark brown tegmen; four apical cells in hind wings.

Description. Body length without tegmen about 11.10 mm in XDB3-1209 (Figure 2A), 11.88 mm in XDB3-0252 (Figure 2G), and 12.03 mm in XDA2-0430 (Figure 2H); tegmen 7.80 mm long and 3.00 mm wide in XDB3-1209, 9.27 mm long and 3.00 mm wide in XDB3-0252, 8.10 mm long and 3.21 wide in XDA2-0430. Hind tibia with one lateral spine, apical spines arranged in 2 rows, with 10–14 spines (Figure 2D and Figure 5B). The following measurements are based on the three specimens.

Coloration (Figure 2A,G,H, Figure 5A,C,D and Figure 6). Body and wings generally dark brown; three longitudinal pale stripes on pronotum and scutellum; legs pale with dark banding; two distinct transverse bands and two pale spots at the base on the corium and clavus on each tegmen. Abdomen dark brown.

Head (Figure 2B and Figure 3C). Head subtriangular, length about 1.17–1.57 mm in the midline, width 2.38–2.68 mm in dorsal view, narrower than pronotum, slightly wider than the anterior margin of scutellum; tylus subquadrate; compound eyes globular, about 0.71 mm wide; two ocelli on crown, divided by the central longitudinal ridge, separated from each other by approximately two ocelli diameter, closer to the compound eyes than the distance between one and the other; pedicel of antennae cylindrical, visible dorsally; frontoclypeus width about twice as long at the widest point, without median carina; labrum subtriangular, about 0.16 mm long, slightly swollen; lora (mandibulary plates) narrow, with upper angles reaching half of the frontocypeus length, lower angles reaching level of anteclypeus; maxillary plates narrow, genae narrow, subocular portions distinct. Rostrum (labium) with distinct longitudinal groove, length about 0.93 mm, width about 0.34 mm.

Thorax (Figure 2 and Figure 3C). Pronotum distinctly wider than head with compound eyes, about 2.26–2.78 mm long in mid line, about 3.63–4.05 mm wide at the widest part, subhexagonal, laterally expanded, without median carination; anterolateral margin straight or slightly concave, diverging posteriad, lateral angles merely apically rounded, at about half of pronotum length; posteriolateral margins converging, slightly concave, posterior margin narrower than anterior margin, with a shallow incision. Mesonotum triangular, elongate, length about 2.00–2.30 mm, width about 1.80 mm at base, about 0.8–0.9 as long as pronotum in midline, without protuberances. Prothoracic leg with procoxa length about 0.99–1.24 mm, width about 0.40 mm; protibia length about 1.15–1.47 mm, width about 0.32 mm; protarsus about 1.00 mm long with claw; tarsal claw asymmetrical; Mesothoracic leg with mesofemur length about 1.76–2.08 mm, width about 0.53 mm; mesotibia nearly as wide as protibia, about 1.2–1.8 times as wide as metatibia. Metathoracic leg (Figure 2D and Figure 5B) with metatibia slender, with one distinct lateral spine, in the distal half of the metatibia, apical teeth arranged in two straight rows, apical row without setae; metatarsus about 1.40–1.70 mm long with claw, basi- and midtarsomere armed with apical teeth, teeth arranged in curved rows, apical tarsomere slender; claw robust, hooked, sharp apically, subungual process not clearly visible.

Tegmen (Figure 2, Figure 4, and Figure 5) about 7.80–9.27 mm long, about 3.00 mm wide; covered with setae; broadest at about half of its length; costal margin curved at base then less curved towards wide anteroapical angle, apex round, posteroapical angle widely rounded, tornus short, claval margin merely arcuate; corium punctate, membrane with sparce, indistinct reticulation, appendix present, narrow. Basal cell about 6 times as long as wide, irregularly urceolate. Veins Pc + CP more distinct at base forming narrow ‘hypocosta’ reaching slightly posteriad of half of the tegmen length. Stem ScP + R + MP + CuA thickened at base, ScP emerged and shifted from common stem basad of half of basal cell length, re-entering stem R at level about half of the post-basal cell common stem MP + CuA. Stem ScP + R forked basad of half of the tegmen length; branch (ScP + RA arcuate, subparallel to anterior margin of tegmen, obsolete in terminal section; branch RP curved at base then subparallel costal margin to slightly shifted mediad, obsolete in terminal section. Short common stalk of MP + CuA closing basal cell slightly oblique; common stem MP + CuA shorter than stem R, forked at about basal ⅓ of tegmen length; stem MP curved and directed mediad, forked at about apical ¼ of tegmen length, basad of apex of clavus, at level of the line separating corium and membrane; branch MP_1+2_ forked very close to apical margin of tegmen, vein ending before reaching margin. Branch CuA curved at base then subparallel to claval line (CuP), forked merely basad of MP forking, basad of claval apex; branch CuA_1_ fused with MP_3 + 4_, evanescent in terminal section, not reaching margin; branch CuA_2_ reaching tornus slightly apicad of claval apex. Vein CuP straight, parallel to claval fold, delimiting clavus. Clavus long, with apex exceeding ¾ of tegmen length, claval vein Pcu slightly sigmoid, reaching A_2_ (claval margin) at distance slightly longer than the distance between apex of claval CuP and apex of CuA_2_; vein A_1_ arcuate, not very distinct, reaching margin at about the level of MP-CuA forking.

Hind wing (Figure 2C and Figure 4B) subtriangular, with wide appendix, delicately shagreened; stem ScP + R forked at about half of hind wing length, stem MP single, sinuate, stem CuA forked slightly apicad of stem ScP + R fork, terminal portions of CuA_1_ and CuA_2_ subparallel; terminal portion of CuP diverging from terminal CuA_2_; visible terminal portion of Pcu arcuate, diverging from CuP. Interradial apical cell triangularly elongate, longer than radial cell; radial cell subtrapezoid, wider near hind wing margin, closed basally with oblique veinlet rp-mp; medial cell long, basal closing with mp-cua; cubital cell elongate, about 1.5–3.5 times as long as wide; distance between terminal points CuA_1_ and CuA_2_ longer than distance between RP and MP as well as MP and CuA_1_.

Abdomen (Figure 2A,E,F) with sternites 3rd to 8th well visible. Male terminalia (XDB3-1209, Figure 2E): pygofer narrow, with expansions exceeding line of genital valve apical margin, genital valve (9th abdominal sternum) fused with pygofer, with median incision at posterior margin; subgenital plates elongated-triangular, not fused with each other, not fused with genital valve; anal tube narrow, exceeding apex of subgenital plates, anal style elongate. Female terminalia (XDB3-0252, Figure 2F): pygofer about half of the preserved gonoplac length; base of gonoplac slightly expanded, apex acutely rounded, reaching the level of the tip of anal style, anal tube subcylindrical and without process; anal style short and rounded.

## 4. Discussion

### 4.1. Taxonomic Remarks

Following the description of Cercopidae by Lallemand [30,31] and Schöbel and Carvalho [32], the new fossil falls in the Cercopidae because of the following characters: head narrower than pronotum, approximately as wide as anterior margin of scutellum; median ocellus absent; compound eyes globose; antennae arising in cavities below anterior margin of head, before the eyes; pronotum hexagonal, wider than long; hind legs with one or two lateral spines and rows of apical spines on the tibiae.

The most widely accepted classification divides Cercopidae into two subfamilies: the Old World Cercopinae and the New World Ischnorhininae (previously known as Tomaspidinae) [33,34], however, the Cercopinae was shown to be a paraphyletic group by the present phylogenetic analyses [3]. Hamilton [34,35] proposed to synonymize Aphrophoridae under Cercopidae, but this opinion is not universally accepted. The Old World Cercopinae is characterized by the either partially fused (in Cosmoscartinae) or entirely free subgenital plates, while the subgenital plates of the New World Ischnorhininae are completely fused to the pygofer [14,33]. In our Tibetan fossil, the subgenital plates are partially fused to the pygofer. Therefore, our fossil is assigned to the Cercopinae.

The fossil resembles the tribe Cosmoscartini mainly for the characters of the inflated and striated frontoclypeus, smoothly shifting from crown to face, without median carination, ocelli at almost the same distance to each other as their distance from the compound eyes and male genitalia with subgenital plates without an apical spine-like process [30,31,36]. Cosmoscartini comprises a large number of brightly colored Old World tropical taxa, including 11 genera [7,36]. Hamilton [8] proposed another grouping of Cosmoscartini, excluding *Phymatostetha*. He placed the latter together with two Neotropical genera in a newly created unit called ‘Phymatostethini’. Unfortunately, Phymatostethini’ must be treated as a *nomen nudum*, as a clear definition and formal taxonomic decision with concept and content were not provided.

The placement of the fossils represented here is ensured by comparison with all the extant genera of the tribe Cosmoscartini. Within Cosmoscartini, the genera *Cosmoscarta*, *Ectemnonotum* and *Gynopygocarta* are excluded because the posterior margins of the pronotum are straight (Figure 3B) [37,38,39], while in our fossil the posterior margin is slightly sinuated. *Ectemnonotops* and *Homalostethus* are different from the cercopid fossil in their rounded posterior margin of the pronotum [38]. The main character used to distinguish *Oxymegaspis* from our fossil is the acute scapular angles of the pronotum [40]. The genus *Kanozata* is excluded due to the comparatively small pronotum, scutellum narrowed in the middle, and tegmen significantly longer than the abdomen [41]. The two characters “scutellum narrowed in the middle” and “a large shallow dimple on its anterior half of scutellum” exclude affinities with *Porpacella* [30]. The shape and ratio of the tegmen, the very long clavus, and the lack of wrinkles on the lateral margins of pronotum exclude its placement in the genus *Neoporpacella*. In genus *Gynopygoplax* the CuA of the hind wing forks basad of the cross vein rp-mp, which is different from the fossil [31,42]. Some of the morphological characters of the newly described fossil place it close to *Phymatostetha*: ocelli slightly closer to each other than the compound eyes; pronotum with posterior margin and posterior lateral margins sinuated, scapular angles of the pronotum rounded; scutellum evenly tapering; legs of moderate length, posterior tibiae armed with one spine [31,43]. However the following combination of characters places the fossil in a separate entity within Cosmoscartini, as this tribe is currently understood: angulate border angulate border of frontoclypeus in dorsal view, anterior margin of pronotum protruded, posterior margin with distinct incision, long mesonotum (distinctly longer than half of the length of the pronotum in midline), more basal forking of ScP + R, long clavus, short tornus, and subtriangular shape of subgenital plates place the fossil separately within Cosmoscartini as this tribe is currently understood. Therefore, we describe a new genus and a new species *Nangamostethos tibetense* gen. et sp. nov. based on the unique characters mentioned above.

### 4.2. Biogeographic Implications

There is a large number of fossil record of Cercopoidea, comprising taxa from the Lower Jurassic to the Pleistocene [4,42,43,44]. However, the vast majority of taxa described in the 19th and 20th centuries are in need of urgent revision. Reliable record of Cercopidae is more limited, including a fossil from Ping Chau Island, Hong Kong, of age still disputable—Paleocene [11] or Lower Cretaceous (Berriasian-Valanginian) [43,45]; another reliable cercopid record reported from Paleocene/Eocene deposits of Isle of Mull, Scotland [46]; an inclusion in the Eocene Baltic amber [47]; and another imprint from terminal Eocene of Isle of Wight [48]. Any of these fossils are placed according to current tribal classification, and they are represented by isolated tegmina. Currently, the family Cercopidae is subdivided in two subfamilies, with 17 tribes, comprising about 170 genera [7]. The New World Ischnorhininae were recognized as a monophyletic unit [49], while monophyly of the Cercopinae should be tested [50]. The fossil record of Cercopidae from Asia is rare (Figure 7), one record from Paleocene deposits from southern China [11], one from the Eocene deposit of the Tibetan Plateau reported in this study, and two doubtful taxa from Miocene deposits of Kudia River, Primorski Krai, Russia [51,52]. Weak knowledge of previous fossil records and their phylogenetic relationships [3,50] precluded presentation of detailed models of their evolutionary scenarios. However, the findings of Cercopinae in the Eocene deposits of the Tibetan Plateau described herein gave a new insight on the complex history of the group.

Modern Cercopinae are distributed from temperate to tropical zones of the Old World, being present in a wide array of habitats, from open to dense forests. The Tibetan fossil coexists with a relatively high diversity grass community with palms and numerous other woody taxa, namely a subtropical open woodland ecosystem, with a dry bulb mean annual temperature of ~15.6 °C [53]. The Lunpola–Nima sediment depo-center, placed between the paleo-Gangdese and the paleo-Qiangtang (Tangulha) mountains [54,55] could be one of the areas of modern lineage diversification and a source of current diversity and disparity of the Cosmoscartini. Subsequent paleoclimatic changes [27,56,57], possibly as a response to Tibetan landscape evolution, affected insect population dynamics, and their geographic distribution. As ectotherms, insects are highly sensitive to ambient temperatures and may respond very quickly to temperature fluctuations [58,59]. The fossil cercopid described above, *Nangamostethos* gen. nov., inhabited the TP and possibly became extinct due to regional aridification and an overturn of plant taxa in the late Paleogene [53], whereas its modern congeners are widely distributed and diversified in South-Eastern Asia.

More phylogenetic investigations and an increase in knowledge and elaboration of fossil materials regarding spittlebugs are needed for a better understanding of the origin and evolution of the tribes and families of the superfamily Cercopoidea.

## Figures and Tables

**Figure 1 insects-13-00770-f001:**
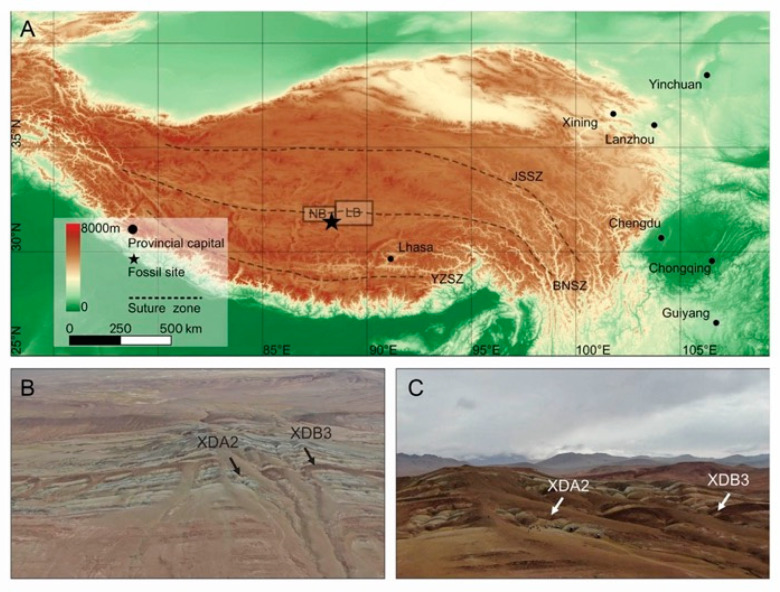
Maps showing the location of the fossil specimens in this study. (**A**) Localities of the Nima Basin and the adjacent Lunpola Basin in the central Tibetan Plateau. BNSZ, Bangong–Nujiang suture zone; JSSZ, Jinshajiang suture zone; LB, Lunpola Basin; NB, Nima Basin; YZSZ, Yalu–Zangbo suture zone. Map data provided by SRTM data V4 [18]. (**B**,**C**) The specimens were collected from fossil-bearing layers XDA2 and XDB3 in the Nima Basin, which are deposited in the same strata.

**Figure 2 insects-13-00770-f002:**
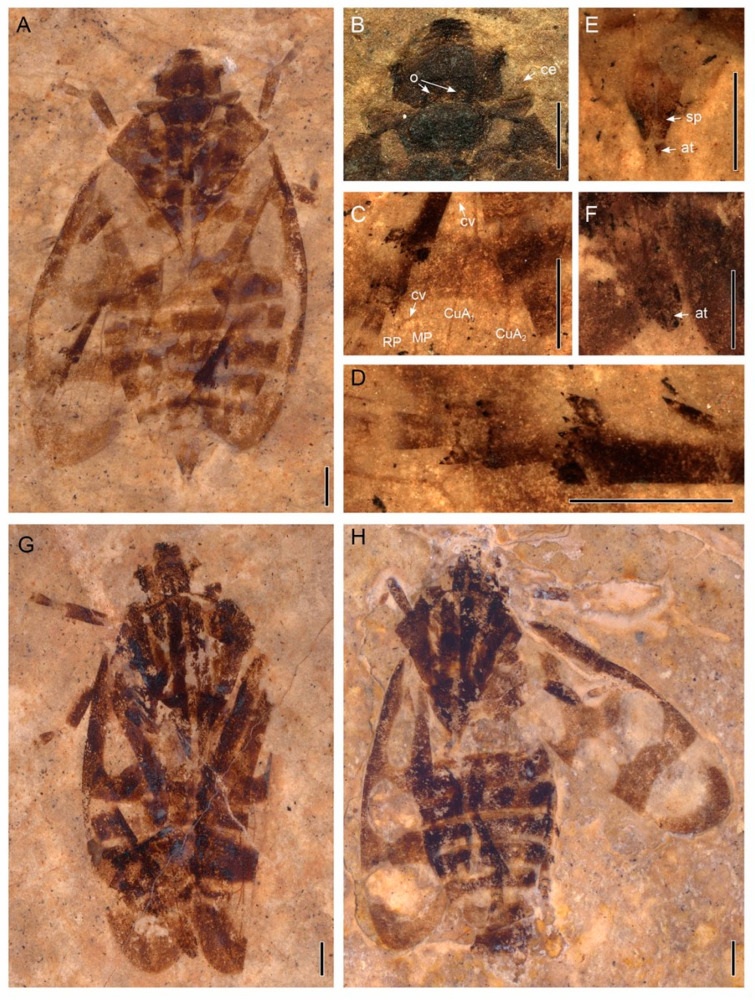
Photographs of the *Nangamostethos tibetense* gen. et sp. nov. (**A**) Body, XDB3-1209, holotype, male. (**B**) Enlargement of (**A**), showing details of the head. (**C**) Enlargement of the hind wing of XDB3-1209, showing details of the veins. (**D**) Enlargement of A, showing the details of the tip of the hind leg. (**E**) Enlargement of (**A**), showing the details of the male terminalia. (**F**) Enlargement of (**G**), showing the details of the female terminalia. (**G**) Body, XDB3-0252A, paratype, female. (**H**) Body, XDA2-0430A, paratype. At, anal tube; ce, compound eye; CuA, cubital anterior; cv, cross vein; MP, media posterior; o, ocelli; RP, radius posterior; sp, subgenital plate. The specimens were immersed into alcohol except in (**B**). Scale bars: 1 mm.

**Figure 3 insects-13-00770-f003:**
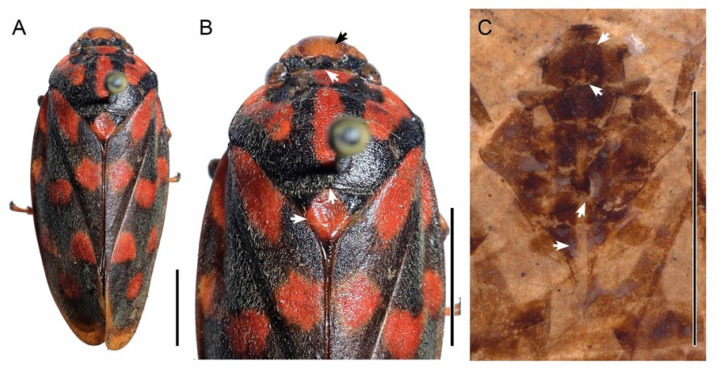
*Nangamostethos tibetense* sp. nov. and some modern Cercopidae. (**A**,**B**) *Phymatostetha dorsivitta*, collection of Kunming Institute of Zoology (Collected from Xinping, Yuxi City, Yunnan Province, China on 12 August, 1978. Collector not recorded). (**C**) Head and thorax of XDB3-1209. The arrows in (**B**,**C**) indicate the differences between the fossil and *Phymatostetha.* Scale bars: 5 mm.

**Figure 4 insects-13-00770-f004:**
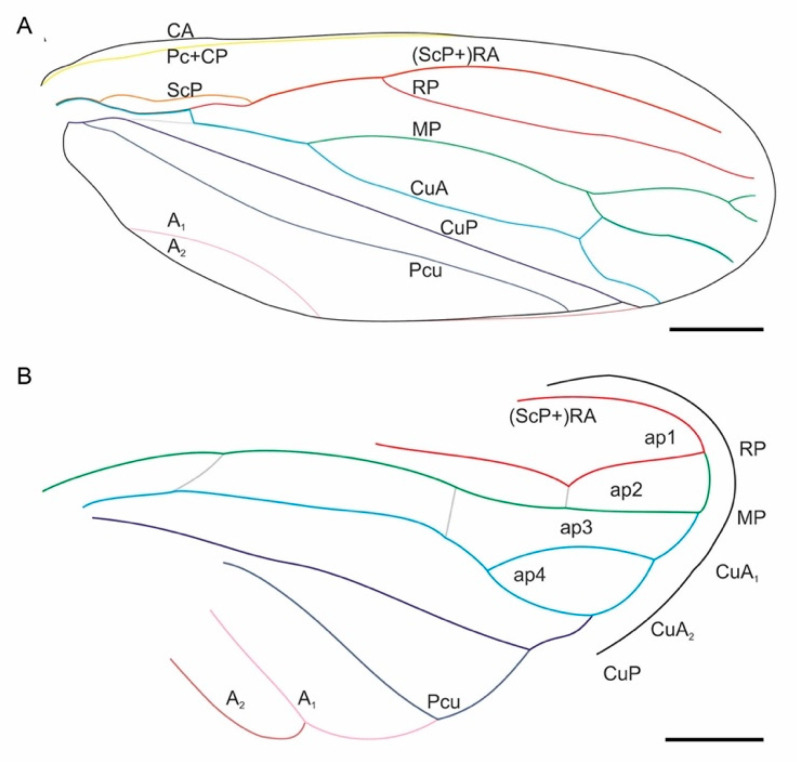
Line drawings of wing venation of *Nangamostethos tibetense* gen. et sp. nov. (**A**) Line drawing of the tegmen of XDA2-0430A. (**B**) Interpretive drawing of the hind wing based on XDB3-1209. A_1_, first anal vein; A_2_, second anal vein; ap1, interradial apical cell; ap2, radial cell; ap3, medial cell; ap4, cubital cell; CA, costa anterior; CP, costa posterior; CuA, cubital anterior; CuP, cubitus posterior; MP, media posterior; Pc, precosta; Pcu, postcubitus; RA, radius anterior; RP, radius posterior; ScP, subcosta posterior. Scale bars: 1 mm.

**Figure 5 insects-13-00770-f005:**
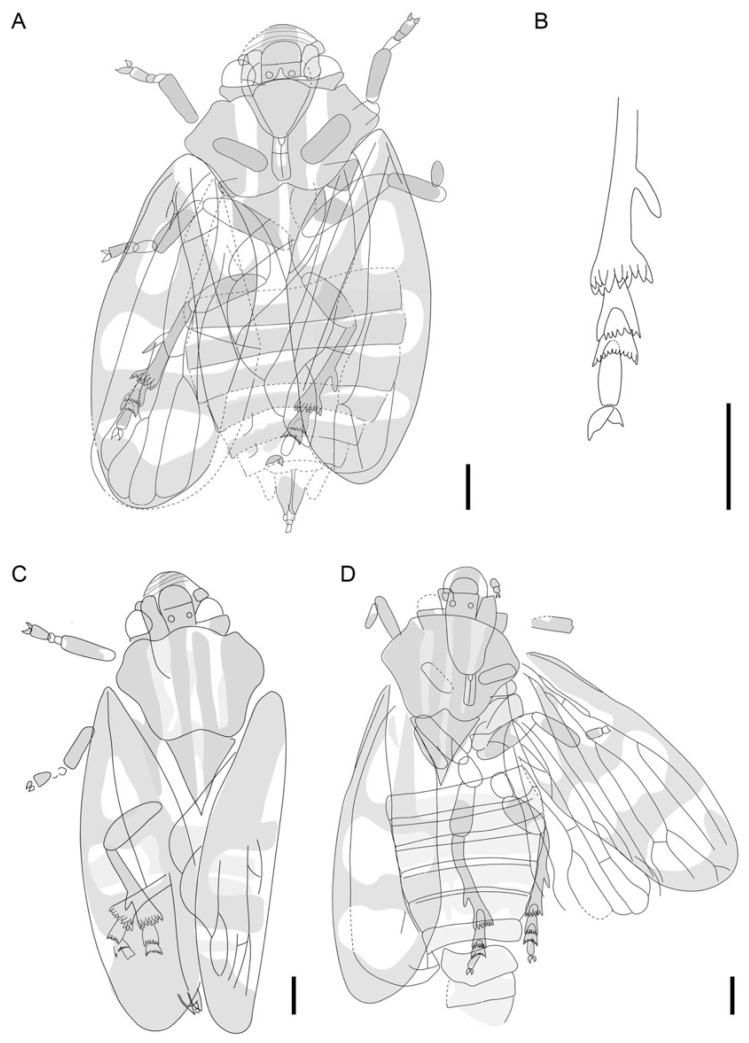
Line drawings of *Nangamostethos tibetense* gen. et sp. nov. (**A**) General habitus of XDB3-1209. (**B**) Interpretive drawing of the hind leg based on XDB3-1209. (**C**) General habitus of XDB3-0252A. (**D**) General habitus of XDA2-0430A. Scale bars: 1 mm.

**Figure 6 insects-13-00770-f006:**
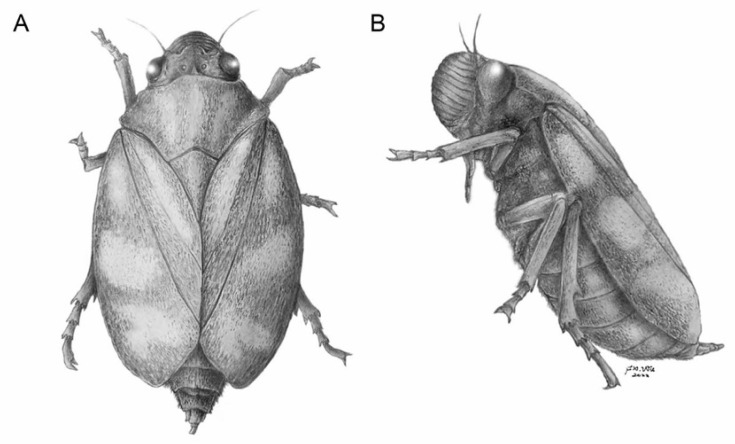
Reconstruction of *Nangamostethos tibetense* gen. et sp. nov. (**A**) Dorsal view. (**B**) Lateral view. Art by Fei-Xiang Wu.

**Figure 7 insects-13-00770-f007:**
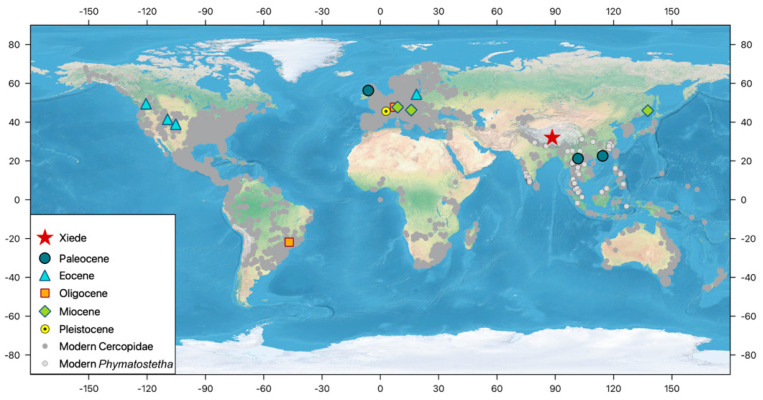
Localities of fossil taxa ascribed to Cercopidae and the modern relatives (note that the taxonomic placement of the fossil record presented here is in need of revision). Localities of fossil sites according to Fossilworks.org [9] and PaleoBioDB.org [10]; modern distribution of *Phymatostetha* and Cercopidae according to GBIF.org (doi: https://doi.org/10.15468/dl.desftn; https://doi.org/10.15468/dl.fcfwj6 (accessed on 14 April 2022)); map data provided by Natural Earth II.

## Data Availability

All the data used in this study are presented in the manuscript and figures.

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
