# Peer review of "A New Genus of Spittlebugs (Hemiptera, Cercopidae) from the Eocene of Central Tibetan Plateauâ€"

_insects, 2022, doi:10.3390/insects13090770_

Round 1
Reviewer 1 Report
Manuscript ID: insects-1871381
A new genus of spittlebugs (Hemiptera, Cercopidae) from the Eocene of central Tibetan Plateau
The authors describe a new genus and a new fossil species Nangamosthetha gen. nov and N. tibetensis sp. nov. from the late Eocene of the central Tibetan Plateau, China. They provide a detailed description with good photographs and line drawings and make an extensive comparison with extant genera of the same tribe, Cosmoscartini. This new species represents one of the few (4?) fossils of Cercopidae from Asia and the first Eocene fossil from this region. They hypothesize that this species may have become extinct due to aridification and vegetation disturbance in the region. The work is interesting, valuable and suitable for publication in Insects.
My comments
Summary
L36-37. Substitute "few" for the actual number of fossils.
Introduction
L-49-50. The introduction does not mention how many species have been described so far, the geographic areas, or the geologic epoch. Some of this information is found in the discussion: L258-262 and L307-321. It would be useful to have a brief description of the current knowledge of the fossils of the family. A table with the subfamilies/tribes with fossil taxa indicating geological epoch, distribution, and current status (reliable/unreliable) would help the readers understand the background and relevance of this work.
Results
L. 101. Why systematic paleontology? There are not two different kinds of systematics.
Figure 6. The caption is not clear. It says it represents the distribution of fossils and fossil sites. Does each symbol represent only one species? The word distribution should be used only to refer to the distribution of taxa, but not to the locations where fossils have been found. Also, the map in Figure (A) shows that the fossils have been found in a red-colored area which, according to the caption, this color represents the 8000m altitude however, in the text, it is said that the fossils were collected at 4662 m altitude. Please check.
L.110. The authors refer to the transliteration of the word στήθος as 'stethos' but they named the genus Nangamosthetha, when it should be Nangamostetha" without "h". In modern Greek, the gender of the word στήθος is neuter, το στήθος, not masculine. According to ICZN, when a gender noun is a compound word, the gender is given by the final component (article 30.1.1). Thus, it should end in "stethos". Please check the declension of Tibet, “tibetense” is the neutral adjective.
L-125-126. Do these measurements apply to all species of a genus? If not, they are best left as proportions.
L-155. Remove “face with”
L-181. Poteroapical?
L186-188. ScP shifted basad of half of basal cell length and forked basad of half of tegmen length. Please clarify.
L187-197. The content of these two long sentences needs to be separated by several semicolons (;)
L 231. Figure 3. Dorsivitta. Correct.
L232. Use capitals: Kunming Institute of Zoology
L438. Remove 3335
L 442. Add a space before “fur”
L472. Ass a space before 12th
L-482. Remove &
Author Response
Response to Reviewer 1 Comments
Point 1: L36-37. Substitute "few" for the actual number of fossils.
Response 1: As indicated in L. 321–324 of the revision, the Cercopidae fossil record located in Asia including one record from Paleocene deposits from southern China, one from Eocene deposit of the Tibetan Plateau reported in this study, and two doubtful taxa from Miocene deposits of Russia. The actual number of fossils located in Asia is not clear for now.
Point 2: L-49-50. The introduction does not mention how many species have been described so far, the geographic areas, or the geologic epoch. Some of this information is found in the discussion: L258-262 and L307-321. It would be useful to have a brief description of the current knowledge of the fossils of the family. A table with the subfamilies/tribes with fossil taxa indicating geological epoch, distribution, and current status (reliable/unreliable) would help the readers understand the background and relevance of this work.
Response 2: We have added a brief description of the fossil record of Cercopidae in L. 55–59 of the revision: “Fossil record of Cercopidae is large, with more than forty species reported from the Paleocene to the Pleistocene in the Eurasia and Americas [9,10]. However, most of the record was attributed to the genera or even the family with relatively weak support. The reliable record of Cercopidae is more limited, and therefore need a further revision.” The current knowledge of the fossils of the family is so small that we do not think it is reasonable to put just a table of the list, without detailed comment on each taxon. The revision of those fossils will be another large paper one of our authors going to work on.
Point 3: L. 101. Why systematic paleontology? There are not two different kinds of systematics.
Response 3: We have deleted the “paleontology” in L. 104 of the revised version.
Point 4: Figure 6. The caption is not clear. It says it represents the distribution of fossils and fossil sites. Does each symbol represent only one species? The word distribution should be used only to refer to the distribution of taxa, but not to the locations where fossils have been found. Also, the map in Figure (A) shows that the fossils have been found in a red-colored area which, according to the caption, this color represents the 8000m altitude however, in the text, it is said that the fossils were collected at 4662 m altitude. Please check.
Response 4: We have changed the word “distribution” into “localities” in L. 305–306, the caption of Figure 7. Some symbols represent more than one species, whereas the exact number need further revision. Besides, we have revised the Figure 1A, increasing the contrast of the color of the bar.
Point 5: L.110. The authors refer to the transliteration of the word στήθος as 'stethos' but they named the genus Nangamosthetha, when it should be Nangamostetha" without "h". In modern Greek, the gender of the word στήθος is neuter, το στήθος, not masculine. According to ICZN, when a gender noun is a compound word, the gender is given by the final component (article 30.1.1). Thus, it should end in "stethos". Please check the declension of Tibet, “tibetense” is the neutral adjective.
Response 5: We have corrected the name as “Nangamostethos tibetense”.
Point 6: L-125-126. Do these measurements apply to all species of a genus? If not, they are best left as proportions.
Response 6: Yes. There is only one species in the new genus Nangamostethos.
Point 7: L-155. Remove “face with”
Response 7: We have deleted the “face with”.
Point 8: L-181. Poteroapical?
Response 8: We have corrected the word as “posteroapical”.
Point 9: L186-188. ScP shifted basad of half of basal cell length and forked basad of half of tegmen length. Please clarify.
Response 9:
The venational patterns and interpretation in Cicadomorpha and Cerrcopoidea is still subject of vivid discussions, however (work in progress; subject of another paper in preparation) some general patterns interpreted from the Hemiptera general pattern could be taken in interpretation. The costal compex of veins is composed of three main stems precosta, costa anterior and cota posterior, these are tightly fused in Cercopoidea however in basal portion of Pc (and its fusion with CP) could be shifted forming the structure sometimes called hypocostal plate, hypocostal area, hypocosta. The basal stem composed of ScP+R+MP+CuA (as inferred from general hemipteran pattern – however, there are discussion on this interpretation) is starting from a common point; vein ScP in Cercopoidea quite early is separating from it, emerging from this main stem and shifted for a distance, running separately, then, afer separation of MP+CuA from common, composed stem (R+MP+CuA) it fused again with main stem of R. Such pattern is very characterisitc of Cercopoidea, however more or less expressed independence of ScP is found among other hemipterans as well.
This section is reworded in MS file.
Point 10: L187-197. The content of these two long sentences needs to be separated by several semicolons (;)
Response 10: We have separated the long sentences using semicolons. Please see L. 190–200 of the revision.
Point 11: L 231. Figure 3. Dorsivitta. Correct.
Response 11: We have corrected the name as “Phymatostetha dorsivitta” in L. 234–235.
Point 12:
L232. Use capitals: Kunming Institute of Zoology
L438. Remove 3335
L 442. Add a space before “fur”
L472. Ass a space before 12th
L-482. Remove &
Response 12: We have corrected these mistakes accordingly.
Reviewer 2 Report
This is a well-researched and well-prepared manuscript reporting a new extinct spittlebug taxon; no edits recommended from this reviewer.
Author Response
There is no comment needed to reply according to the reviewer.
Reviewer 3 Report
A new genus of spittlebugs (Hemiptera, Cercopidae) from the Eocene of central Tibetan Plateau
by Xiao-Ting Xu, Jacek Szwedo, Di-Ying Huang, Wei-Yu-Dong Deng, Martyna Obroślak, Fei-Xiang Wu, and Tao Su
The manuscript describes a new fossil genus and species from the central Tibetan Plateau. Detailed descriptions, photographs and line drawings of the new taxon are presented, even including a reconstruction of what the species may have looked like in real life. Congratulations to the artistic skills of the author alongside his scientific skills. Justification for its placement in Cosmoscartini is provided as well as reasons why the new species does not fit into any existing genera and required the creation of a monotypic genus. Possible causes of its extinction are discussed. The paper represents a valuable contribution to our knowledge of the Cercopidae, i.e. since it is one of the few fossils in this family known from Asia and I highly recommend it for publication in Insects.
1) Lines 15-16: Is it necessary to have Chinese Academy of Sciences in both lines when talking about the same laboratory?
2) Line 22: change to: … commonly named “spittlebugs” …
3) Line 23: change to: … produce a spittle mass …
4) Line 23: not sure if ‘defend’ is the ideal word. Better wordings might be: nymphs hide from predators and parasites in masses of spittle they excrete. Other wordings could be: camouflage; spittle acts as a deterrent to predators and parasites
5) Lines 24 and 33, 273: 11 genera are listed for Cosmoscartini, which I would assume are the ones listed in the text between lines 273 and 290. What about the following genera, are they still in Cosmoscartini?
Opistharsostethus Schmidt 1911: moved into Cosmoscartini by Lallemand & Synave 1961
Leptataspis Schmidt 1910: listed under Cosmoscartini by Hamilton 2016
Kotozata Matsumura 1940: listed under Cosmoscartini by Dimitriev (dimitriev.speciesfile.org)
Whilst the genera of Cosmoscartini are mentioned in the text, summarising them in a table as a checklist (including information on their distribution) could be a very useful addition to the paper.
6) Line 28: change to: … represents one of few fossils …
7) Line 29: change to: … extinct from the TP due to regional aridification …
8) Line 31-38: above comments from the Simple Summary also apply to the Abstract
9) Line 38: change to: … plant taxa in the late Paleogene.
10) Line 40: ‘fossil’ may be included in Keywords
11) Line 46: change to: ... on the hind tibia …
12) Line 47: change to: ... world, centres of species diversity are in the tropics …
13) Line 48: change to: … of the described …
14) Line 49: change to: ... extant families (Cercopidae, Aphrophoridae, Epipygidae, Clastopteridae and Machaerotidae) and three extinct ones (Procercopidae, Sinoalidae and Cercopionidae).
15) Line 52: change to: ... attempted [3, 6], which means the monophyly …
16) Line 54: change to: ... which bear …
17) Line 56: add author of the species Allocercopis punctatis
18) Line 57: change to: ... recorded from the Cretaceous …
19) Line 58: change to: ... Tegmen patterns in the subfamily Cercopinae often include red, orange or yellow markings on a black background [12].
20) Line 60: change to: ... coloration, which probably in conjunction with reflex bleeding (exuding hemolymph from rupture lines in pretarsal pads when attacked) acts to deter predators [13].
21) Line 62: change to: ... oligo- or polyphagous …
22) Line 64: A high proportion of cercopids, in regards to abundance and species diversity, show a statistically significant preference for nitrogen-fixing host plants …
23) Line 71: change to: ... in the central Tibetan …
24) Line 74: change to: ... in the Nima Basin …
25) Line 80: change to: The Nima Basin is … in the central Tibetan …
26) Line 85: change to: The Nima Basin …
27) Line 86: Line 80: change to: … to the latest chronostratigraphic …
28) Line 93: change to: … with a Canon 77D digital camera attached …
29) Lines 105 and 128: first names of authors can be deleted
30) Line 112: remove second comma
31) Lines 113 and 148: change to: … base of corium …
32) Throughout the entire document the term ‘pale’ should be used instead of ‘light’, i.e. for markings, e.g. Line 113: … and two pale spots ….
33) Line 112: change to … two distinct pale transverse …
34) Line 114: change to … ScP+R (in Cosmoscarta and Phymatostetha stem ScP+R is forked closer to the midlength of the corium).
35) Line 115: not sure what is meant with ‘more clear section of CuA1’? Maybe a wording like ‘more simplified branching pattern of CuA1’ could be used.
36) Line 116: add space between Phymatostetha and Cosmoscarta
37) Line 125: correct spelling of Phymatostetha
38) Line 137: change to: …scutellum; legs pale with dark banding …
39) Line 144: should that be a dash between 10 and 14, currently looks more like a tilde.
40) to comply with telegraph style in descriptions replace ‘and’ with ‘,’ in Lines 151, 160, 163, 170, 171, 172. Delete ‘the’ in Line 163.
41) Line 168: change to: … 1.80 mm at base …
42) Line 175: change to: … in distal half of metatibia, apical teeth …
43) Line 181: change to: … then less distinctly curved towards wide … … round, posteroapical …
44) Line 183: change to: … membrane with sparse, indistinct reticulation …
45) Line 189: change to: … subparallel to anterior margin of tegmen, obsolete in terminal section …
46) Line 191: shouldn’t that be stem ScP+R instead of just R?
47) Line 192: … MP curved …
48) Line 193: not sure what ‘at line’ means
49) Line 194: MP1+2 forked very close to apical margin of tegmen, vein ending before reaching margin.
50) Line 196: change to: …MP3+4, evanescent in terminal section, not reaching margin …
51) Lines 179-210: Due to very useful photographs and line drawings the curvature and special features of the veins do not need to be described in that much detail, as it can more easily be understood from the illustrations. Just focus on the most important characters, or explain details that are not included or obvious in the illustrations.
52) Line 214: change to: … not fused with each other …
53) Line 218: ‘not concave’ - it may be better to describe what it is instead of what it isn’t. e.g. cylindrical? straight?
54) Line 221: … of Nangamosthetha …
55) Line 222 and following lines: change to: … showing details of the head.
56) Line 223: change to: … showing details of the veins.
57) Line 224: change to: … showing details of the tip of the hind leg … details of the male genitalia …
58) Line 225: change to: … showing details of the female genitalia…
59) Line 231: change to: … and some modern … dorsivitta (Walker, 1851), collection …
60) Line 232 and 362: change to: … Kunming Institute of Zoology
61) Line 232: not sure what the date after Kunming Institute of Zoology stands for, but most likely can be deleted, or if left at least explained what it means.
62) Line 237: change to: … drawings of wing venation …
63) Line 238: : change to: … wing based on …
64) Line 245: : change to: … leg based on …
65) Line 257: change to: … lateral spines …
66) Line 260: change to: … Tomaspidinae) [31, 32], however the Cercopinae were shown to be a paraphyletic group by recent phylogenetic analyses …
67) Line 263: change to: … World Cercopinae are characterized by either … in Cosmoscartini) …
68) Line 268: change to: … resembles the tribe Cosmoscartini …
69) Line 270: change to: … distance to each other … distance from the compound …
70) Line 272: change to: … Cosmoscartini comprise … number of brightly …
71) Line 274: change to: … excluding Phymatostetha. He placed the latter together with two Neotropical genera in a newly created unit called ‘Phymatostethini’. Unfortunately, Phymatostethini’ must be treated as a nomen nudum, as a clear definition and formal taxonomic decision with concept and content …
72) Line 277: after ‘provided’ a new paragraph needs to be started that has some sort of an introductory phrase to tell readers you are now talking about the generic placement of the fossil and no longer about tribal arrangements. Otherwise one might think you are, like Hamilton, trying to exclude certain genera from the tribe Cosmoscartini.
73) Line 278: change to: … are characterised by a straight posterior margin of the pronotum … while in our fossil the posterior margin is slightly sinuate.
74) Line 280: change to: ... cercopid fossil in their …
75) Line 281: change to: … Oxymegaspis from our fossil are the acute scapular angles of the pronotum [38].
76) Line 286: change to: … of the tegmen, the very long clavus and the lack of ... pronotum of the fossil exclude its placement in the …
77) Line 288: change to: … Gynopygoplax CuA of the hind wing forks basad of the crossvein …
78) Line 290: change to: … ocelli slightly closer to each … pronotum with posterior margin and posterior lateral margins sinuated, scapular angles rounded; scutellum evenly tapering; …
79) Line 293: ‘legs moderate’ in which regard? moderate in length? number of spines?
80) Line 294: change to: … However the following combination of characters places the fossil in a separate entity within Cosmoscartini, as this tribe is currently understood: angulate border … long mesonotum (distinctly longer than … pronotum in midline), …
81) Line 299: Therefore, we describe a …
82) Line 307: change to: There is a large number of fossil records of Cercopoidea, comprising taxa from the Lower Jurassic to the Pleistocene … However, the vast majority of taxa described in the 19th and 20th …
83) Line 309: change to: Reliable records of Cercopidae are more limited, including a fossil from …
84) Line 313: “Any of these fossils is placed in tribal classification as recently understood …”. Not sure what the authors intend to say?
85) Line 315: change to: Currently the family Cercopidae …
86) Line 317: change to: … monophyly of the Cercopinae …
87) Line 318: change to: Fossil records of Cercopidae from Asia are rare … one from Eocene deposits …
88) Line 321: change to: Weak knowledge of previous fossil records and their phylogenetic relationships [3, 50] precluded presentation … However, the herein described findings of Cercopinae …
89) Line 326: change to: … in a wide array …
90) Line 326: change to: … Tibetan fossils … or … Tibetan fossil coexists ….
91) Line 327: change to: … a relatively high diversity grass community …
92) Line 329: change to: The Lunpola-Nima …
93) Line 333: change to: … possibly as a response to …
94) Line 336: change to: … inhabited the TP …
95) Line 337: change to: … extinct due to regional … in the late Paleogene …
96) Line 342: change to: … of the tribes and families of the superfamily Cercopoidea.
97) Line 359: change to: … in the Tibetan region …
98) Lines 425 and 427: end references with a full stop “.” instead of “;”
99) Line 438: change number of reference to 35
100) Line 438: “In” should not be in italics.
101) Line 442: add blank spaces between words in journal title
102) Line 451: check reference whether “Öfvers.fauna.” should be deleted.
103) Line 464: change to: … befindlichen Reste der …
104) make sure last accessed dates are given for all ‘website references’
General comments:
· Are there any particular reasons why the authors decided to give the generic name the commonly female Latin ending ‘–a’ when Latinising it and then defining it to be masculine gender? The original Greek word stethos is neuter. Latinising it by turning it into –stethus and calling it a masculine gender would make more sense.
· The gender of the type specimens should be detailed in the Materials section Lines 132-135.
See examples from ICZN Article 30:
30.1.3. a genus-group name that is a Greek word latinized with change of ending, or with a Latin or latinized suffix, takes the gender normally appropriate to the changed ending or the Latin suffix.
Examples. Names with the Latin gender ending -us, latinized from the Greek endings -os (masculine or feminine), -e (feminine), -a (neuter) or -on (neuter), are masculine: e.g. -cephalus (kephale), -cheilus and -chilus (cheilos), -crinus (krinon), -echinus (echinos), -gnathus (gnathos), -rhamphus (rhamphos), -rhynchus (rhynchos), -somus (soma), -stethus (stethos), and -stomus (stoma). Names ending in the Latin gender ending -a, latinized from the Greek ending -on are feminine, e.g. -metopa (metopon). Names derived from the Greek -keras (neuter) may have the ending -cerus (masculine) or -cera (feminine), although simple transliteration of the Greek ending as -ceras retains the neuter gender.
Author Response
Response to Reviewer 3 Comments
1) Lines 15-16: Is it necessary to have Chinese Academy of Sciences in both lines when talking about the same laboratory?
Response: It is part of the Institution name.
Line 22: change to: … commonly named “spittlebugs” …
Response: Changed.
3) Line 23: change to: … produce a spittle mass …
Response: Changed.
4) Line 23: not sure if ‘defend’ is the ideal word. Better wordings might be: nymphs hide from predators and parasites in masses of spittle they excrete. Other wordings could be: camouflage; spittle acts as a deterrent to predators and parasites
Response: The sentence is modified, thanks for the suggestion.
5) Lines 24 and 33, 273: 11 genera are listed for Cosmoscartini, which I would assume are the ones listed in the text between lines 273 and 290. What about the following genera, are they still in Cosmoscartini?
Response: The classification of Cercopinae is still subject of discussions and it is not very stable. New attempts put placement of some taxa in doubt challenging the classical schemes proposed by Lallemand and Synave. Hamilton concept and content proposals for particular taxa are also subjects of discussions. It is one of the reasons that in COOL the tribal classification is not included (Soulier-Perkins, p.c.), while dmitriev.speciesfile database is an aggregation database with listing and attributing of taxa as they were catalogued by Metcalf, with some updates. The paper with a description of a fossil is not an attempt to present a full review or revision of Cosmoscartini or Cercopinae as a whole. This is work in progress and fossils can add new insights into classification and relationships discussions.
Opistharsostethus Schmidt 1911: moved into Cosmoscartini by Lallemand & Synave 1961
Response: Opistharsostethus Schmidt, 1911 – yes it is placed in Cosmoscartini (see Crispolon et al. 2021), the placement also supported by preliminary morphological analyses (Crispolon & Soulier-Perkins 2019). However, the genus was placed in Suracartini by Metcalf (1961).
Leptataspis Schmidt 1910: listed under Cosmoscartini by Hamilton 2016
Response: Leptataspis Schmidt, 1910 - placement in Cosmoscartini is supported by preliminary morphological analyses (Crispolon & Soulier-Perkins 2019), however in a clade with Surascarta and Opistharsostethus. It was placed also in the tribe Euryaulacini (Metcalf 1960).
Kotozata Matsumura 1940: listed under Cosmoscartini by Dimitriev (dimitriev.speciesfile.org)
Response: Kotozata Matsumura, 1940 - yes it is placed in Cosmoscartini (Metcalf 1960), however since its (brief) description, no new morphological or molecular data are available, confirming or denying this placement.
Whilst the genera of Cosmoscartini are mentioned in the text, summarising them in a table as a checklist (including information on their distribution) could be a very useful addition to the paper.
Response: As mentioned above – the paper is not an attempt to present a full review or revision of Cosmoscartini and its concept and content are still subject to controversies. Adding a lengthy list of taxa (not only genera – in many cases the species should be discussed) and review/verification of distributional data it is a subject for separate work. Cercopinae is a group of many taxonomic and nomenclatorial problems, however clarification of these is a work in progress now.
6) Line 28: change to: … represents one of few fossils …
Response: Done, thanks.
7) Line 29: change to: … extinct from the TP due to regional aridification …
Response: Done, thanks.
8) Line 31-38: above comments from the Simple Summary also apply to the Abstract
Response: Done, thanks.
9) Line 38: change to: … plant taxa in the late Paleogene.
Response: Done, thanks.
10) Line 40: ‘fossil’ may be included in Keywords
Response: Done, thanks.
11) Line 46: change to: ... on the hind tibia …
Response: Done, thanks.
12) Line 47: change to: ... world, centres of species diversity are in the tropics …
Response: Done, thanks, corrected to British spelling.
13) Line 48: change to: … of the described …
Response: Done, thanks.
14) Line 49: change to: ... extant families (Cercopidae, Aphrophoridae, Epipygidae, Clastopteridae and Machaerotidae) and three extinct ones (Procercopidae, Sinoalidae and Cercopionidae).
Response: Done, thanks for suggestion.
15) Line 52: change to: ... attempted [3, 6], which means the monophyly …
Response: Done, thanks for suggestion.
16) Line 54: change to: ... which bear …
Response: Done, thanks.
17) Line 56: add author of the species Allocercopis punctatis
Response: Done, thanks.
18) Line 57: change to: ... recorded from the Cretaceous …
Response: Done, thanks.
19) Line 58: change to: ... Tegmen patterns in the subfamily Cercopinae often include red, orange or yellow markings on a black background [12].
Response: Done, thanks for suggestion.
20) Line 60: change to: ... coloration, which probably in conjunction with reflex bleeding (exuding hemolymph from rupture lines in pretarsal pads when attacked) acts to deter predators [13].
Response: Done, thanks.
21) Line 62: change to: ... oligo- or polyphagous …
Response: Done, thanks.
22) Line 64: A high proportion of cercopids, in regards to abundance and species diversity, show a statistically significant preference for nitrogen-fixing host plants …
Response: Done, thanks for suggestion.
23) Line 71: change to: ... in the central Tibetan …
Response: Done, thanks.
24) Line 74: change to: ... in the Nima Basin …
Response: Done, thanks.
25) Line 80: change to: The Nima Basin is … in the central Tibetan …
Response: Done, thanks.
26) Line 85: change to: The Nima Basin …
Response: Done, thanks.
27) Line 86: Line 80: change to: … to the latest chronostratigraphic …
Response: Done, thanks.
28) Line 93: change to: … with a Canon 77D digital camera attached …
Response: Done, thanks.
29) Lines 105 and 128: first names of authors can be deleted
Response: Done.
30) Line 112: remove second comma
Response: Corrected.
31) Lines 113 and 148: change to: … base of corium …
Response: Done.
32) Throughout the entire document the term ‘pale’ should be used instead of ‘light’, i.e. for markings, e.g. Line 113: … and two pale spots ….
Response: Done, thanks for suggestion.
33) Line 112: change to … two distinct pale transverse …
Response: Done.
34) Line 114: change to … ScP+R (in Cosmoscarta and Phymatostetha stem ScP+R is forked closer to the midlength of the corium).
Response: Done.
35) Line 115: not sure what is meant with ‘more clear section of CuA1’? Maybe a wording like ‘more simplified branching pattern of CuA1’ could be used.
Response: Reworded, thanks for suggestion. It is part of CuA1 branching, section of this branch.
36) Line 116: add space between Phymatostetha and Cosmoscarta
Response: Done.
37) Line 125: correct spelling of Phymatostetha
Response: Done.
38) Line 137: change to: …scutellum; legs pale with dark banding …
Response: Done, thanks for suggestion.
39) Line 144: should that be a dash between 10 and 14, currently looks more like a tilde.
Response: Checked and corrected.
40) to comply with telegraph style in descriptions replace ‘and’ with ‘,’ in Lines 151, 160, 163, 170, 171, 172. Delete ‘the’ in Line 163.
Response: Done, thanks for suggestion.
41) Line 168: change to: … 1.80 mm at base …
Response: Done.
42) Line 175: change to: … in distal half of metatibia, apical teeth …
Response: Done, thanks for suggestion.
43) Line 181: change to: … then less distinctly curved towards wide … … round, posteroapical …
Response: Done, thanks for suggestion.
44) Line 183: change to: … membrane with sparse, indistinct reticulation …
Response: Done, thanks for suggestion.
45) Line 189: change to: … subparallel to anterior margin of tegmen, obsolete in terminal section …
Response: Done, thanks for suggestion.
46) Line 191: shouldn’t that be stem ScP+R instead of just R?
Response: Checked, the ScP is partly freed then re-enters stem R.
47) Line 192: … MP curved …
Response: Done.
48) Line 193: not sure what ‘at line’ means
Response: Here it means at line (flexion line, nodal line) separating basal (corium) section of tegmen from more flexible (membrane) terminal portion – these two parts could differ in level of sclerotization, patterns and thickenss of veins and veinletets, sometimes in coloration. It is usualyyan ‘imaginative’ line between apex of clavus and terminal ScP+RA1. Reworded.
49) Line 194: MP1+2 forked very close to apical margin of tegmen, vein ending before reaching margin.
Response: Done, thanks for suggestion.
50) Line 196: change to: …MP3+4, evanescent in terminal section, not reaching margin …
Response: Done, thanks for suggestion.
51) Lines 179-210: Due to very useful photographs and line drawings the curvature and special features of the veins do not need to be described in that much detail, as it can more easily be understood from the illustrations. Just focus on the most important characters, or explain details that are not included or obvious in the illustrations.
Response: Thanks, the drawings are updated and we hope these correspond well with detailed descriptions which are often lacking for the Cercopidae, so we would like to present a kind of standard.
52) Line 214: change to: … not fused with each other …
Response: Done.
53) Line 218: ‘not concave’ - it may be better to describe what it is instead of what it isn’t. e.g. cylindrical? straight?
Response: Reworded, thanks.
54) Line 221: … of Nangamosthetha …
Response: Checked and corrected.
55) Line 222 and following lines: change to: … showing details of the head.
Response: Checked and corrected.
56) Line 223: change to: … showing details of the veins.
Response: Checked and corrected.
57) Line 224: change to: … showing details of the tip of the hind leg … details of the male genitalia …
Response: Checked and corrected.
58) Line 225: change to: … showing details of the female genitalia…
Response: Checked and corrected.
59) Line 231: change to: … and some modern … dorsivitta (Walker, 1851), collection …
Response: Checked and corrected.
60) Line 232 and 362: change to: … Kunming Institute of Zoology
Response: Checked and corrected.
61) Line 232: not sure what the date after Kunming Institute of Zoology stands for, but most likely can be deleted, or if left at least explained what it means.
Response: It is the collecting date and the collection number. Reworded.
62) Line 237: change to: … drawings of wing venation …
Response: Done.
63) Line 238: : change to: … wing based on …
Response: Done.
64) Line 245: : change to: … leg based on …
Response: Done.
65) Line 257: change to: … lateral spines …
Response: Done.
66) Line 260: change to: … Tomaspidinae) [31, 32], however the Cercopinae were shown to be a paraphyletic group by recent phylogenetic analyses …
Response: Done, thanks for suggestion.
67) Line 263: change to: … World Cercopinae are characterized by either … in Cosmoscartini) …
Response: Done, thanks for suggestion.
68) Line 268: change to: … resembles the tribe Cosmoscartini …
Response: Done, thanks for suggestion.
69) Line 270: change to: … distance to each other … distance from the compound …
Response: Done, thanks for suggestion.
70) Line 272: change to: … Cosmoscartini comprise … number of brightly …
Response: Done, thanks for suggestion.
71) Line 274: change to: … excluding Phymatostetha. He placed the latter together with two Neotropical genera in a newly created unit called ‘Phymatostethini’. Unfortunately, Phymatostethini’ must be treated as a nomen nudum, as a clear definition and formal taxonomic decision with concept and content …
Response: Reworded, thanks for suggestion.
72) Line 277: after ‘provided’ a new paragraph needs to be started that has some sort of an introductory phrase to tell readers you are now talking about the generic placement of the fossil and no longer about tribal arrangements. Otherwise one might think you are, like Hamilton, trying to exclude certain genera from the tribe Cosmoscartini.
Response: Done, thanks for suggestion.
73) Line 278: change to: … are characterised by a straight posterior margin of the pronotum … while in our fossil the posterior margin is slightly sinuate.
Response: Done, thanks for suggestion.
74) Line 280: change to: ... cercopid fossil in their …
Response: Done, thanks for suggestion.
75) Line 281: change to: … Oxymegaspis from our fossil are the acute scapular angles of the pronotum [38].
Response: Done, thanks for suggestion.
76) Line 286: change to: … of the tegmen, the very long clavus and the lack of ... pronotum of the fossil exclude its placement in the …
Response: Reworded, thanks for suggestion.
77) Line 288: change to: … Gynopygoplax CuA of the hind wing forks basad of the crossvein …
Response: Reworded, thanks for suggestion.
78) Line 290: change to: … ocelli slightly closer to each … pronotum with posterior margin and posterior lateral margins sinuated, scapular angles rounded; scutellum evenly tapering; …
Response: Reworded, thanks for suggestion.
79) Line 293: ‘legs moderate’ in which regard? moderate in length? number of spines?
Response: Checked and corrected.
80) Line 294: change to: … However the following combination of characters places the fossil in a separate entity within Cosmoscartini, as this tribe is currently understood: angulate border … long mesonotum (distinctly longer than … pronotum in midline), …
Response: Reworded, thanks for suggestion.
81) Line 299: Therefore, we describe a …
Response: Checked and corrected.
82) Line 307: change to: There is a large number of fossil records of Cercopoidea, comprising taxa from the Lower Jurassic to the Pleistocene … However, the vast majority of taxa described in the 19th and 20th …
Response: Done, thanks for suggestion.
83) Line 309: change to: Reliable records of Cercopidae are more limited, including a fossil from …
Response: Reworded, thanks for suggestion.
84) Line 313: “Any of these fossils is placed in tribal classification as recently understood …”. Not sure what the authors intend to say?
Response: Reworded.
85) Line 315: change to: Currently the family Cercopidae …
Response: Done.
86) Line 317: change to: … monophyly of the Cercopinae …
Response: Done.
87) Line 318: change to: Fossil records of Cercopidae from Asia are rare … one from Eocene deposits …
Response: Done, thanks for suggestion.
88) Line 321: change to: Weak knowledge of previous fossil records and their phylogenetic relationships [3, 50] precluded presentation … However, the herein described findings of Cercopinae …
Response: Done, thanks for suggestion.
89) Line 326: change to: … in a wide array …
Response: Done.
90) Line 326: change to: … Tibetan fossils … or … Tibetan fossil coexists ….
Response: Done.
91) Line 327: change to: … a relatively high diversity grass community …
Response: Done.
92) Line 329: change to: The Lunpola-Nima …
Response: Done.
93) Line 333: change to: … possibly as a response to …
Response: Done.
94) Line 336: change to: … inhabited the TP …
Response: Done.
95) Line 337: change to: … extinct due to regional … in the late Paleogene …
Response: Done.
96) Line 342: change to: … of the tribes and families of the superfamily Cercopoidea.
Response: Done.
97) Line 359: change to: … in the Tibetan region …
Response: Done.
98) Lines 425 and 427: end references with a full stop “.” instead of “;”
Response: Checked and corrected.
99) Line 438: change number of reference to 35
Response: Checked and corrected.
100) Line 438: “In” should not be in italics.
Response: Checked and corrected.
101) Line 442: add blank spaces between words in journal title
Response: Checked and corrected.
102) Line 451: check reference whether “Öfvers.fauna.” should be deleted.
Response: Checked and corrected.
103) Line 464: change to: … befindlichen Reste der …
Response: Checked and corrected.
104) make sure last accessed dates are given for all ‘website references’
Response: Checked and corrected.
General comments:
Are there any particular reasons why the authors decided to give the generic name the commonly female Latin ending ‘–a’ when Latinising it and then defining it to be masculine gender? The original Greek word stethos is neuter. Latinising it by turning it into –stethus and calling it a masculine gender would make more sense.
The gender of the type specimens should be detailed in the Materials section Lines 132-135.
See examples from ICZN Article 30:
30.1.3. a genus-group name that is a Greek word latinized with change of ending, or with a Latin or latinized suffix, takes the gender normally appropriate to the changed ending or the Latin suffix.
Examples. Names with the Latin gender ending -us, latinized from the Greek endings -os (masculine or feminine), -e (feminine), -a (neuter) or -on (neuter), are masculine: e.g. -cephalus (kephale), -cheilus and -chilus (cheilos), -crinus (krinon), -echinus (echinos), -gnathus (gnathos), -rhamphus (rhamphos), -rhynchus (rhynchos), -somus (soma), -stethus (stethos), and -stomus (stoma). Names ending in the Latin gender ending -a, latinized from the Greek ending -on are feminine, e.g. -metopa (metopon). Names derived from the Greek -keras (neuter) may have the ending -cerus (masculine) or -cera (feminine), although simple transliteration of the Greek ending as -ceras retains the neuter gender.
Response: Checked and corrected.